# Bactericidal Permeability-Increasing Protein (BPI) Inhibits *Mycobacterium tuberculosis* Growth

**DOI:** 10.3390/biom14040475

**Published:** 2024-04-13

**Authors:** Silvia Guzmán-Beltrán, Esmeralda Juárez, Brenda L. Cruz-Muñoz, Cesar A. Páez-Cisneros, Carmen Sarabia, Yolanda González

**Affiliations:** Department of Microbiology, National Institute for Respiratory Diseases Ismael Cosio Villegas, Mexico City 14080, Mexico; ejuarez@iner.gob.mx (E.J.); breencroix@gmail.com (B.L.C.-M.); capc031099@gmail.com (C.A.P.-C.); carmen.sarabia@iner.gob.mx (C.S.); ygonzalezh@iner.gob.mx (Y.G.)

**Keywords:** bactericidal permeability-increasing protein, antimicrobial activity, *Mycobacterium tuberculosis*

## Abstract

Bactericidal permeability-increasing protein (BPI) is a multifunctional cationic protein produced by neutrophils, eosinophils, fibroblasts, and macrophages with antibacterial anti-inflammatory properties. In the context of Gram-negative infection, BPI kills bacteria, neutralizes the endotoxic activity of lipopolysaccharides (LPSs), and, thus, avoids immune hyperactivation. Interestingly, BPI increases in patients with Gram-positive meningitis, interacts with lipopeptides and lipoteichoic acids of Gram-positive bacteria, and significantly enhances the immune response in peripheral blood mononuclear cells. We evaluated the antimycobacterial and immunoregulatory properties of BPI in human macrophages infected with *Mycobacterium tuberculosis*. Our results showed that recombinant BPI entered macrophages, significantly reduced the intracellular growth of *M. tuberculosis*, and inhibited the production of the proinflammatory cytokine tumor necrosis factor-alpha (TNF-α). Furthermore, BPI decreased bacterial growth directly in vitro. These data suggest that BPI has direct and indirect bactericidal effects inhibiting bacterial growth and potentiating the immune response in human macrophages and support that this new protein’s broad-spectrum antibacterial activity has the potential for fighting tuberculosis.

## 1. Introduction

Bactericidal permeability-increasing protein (BPI) is an antimicrobial protein capable of killing Gram-negative bacteria [1,2]. BPI acts as an opsonin, facilitating phagocytosis and contributing to the presentation of antigens to immune cells, such as macrophages and dendritic cells. The opsonizing capacity is attributed to the N- and C-terminal portions of BPI, which bridge Gram-negative bacteria and phagocytes [3]. BPI also exerts direct bactericidal activity by forming pores in the membrane, causing the loss of the proton-motive force [3,4,5]. Furthermore, it has been proposed that BPI binds lipopolysaccharides (LPSs) through hydrophobic interactions with the acyl groups, altering their structure, allowing the entry of granulysins and granzymes, and favoring bacterial lysis and death [6,7,8,9]. The location of amino acid 216 near the lipophilic binding pocket in the N-terminal domain suggests a potential binding site for the negatively charged core of an LPS [10]. The C-terminal region contributes to the interaction between BPI and LPSs after the detachment of LPSs from the bacterial envelope [11].

BPI is an anti-inflammatory modulator because it acts antagonistically by stabilizing LPS aggregates and inhibiting the lipopolysaccharide-binding protein (LBP) activity. LBP facilitates the binding of a series of phosphatidylinosides and phosphatidylserine to membrane-bound CD14 receptors [12], resulting in a stimulation of the LPS-induced response in monocytes to generate TNF-α production. The BPI’s function is to join LPSs and decrease the LBP–CD14 engagements, thus blocking their internalization by TLR4. LBP–LPS interaction is required for LPS recognition by TLR4 and MD-2 to induce the inflammatory response [7,13]. BPI has a potent LPS-neutralizing activity because it has a much higher affinity for LPSs than LBP; even at substoichiometric levels, BPI competes with LBP for LPSs [14].

The role of BPI has been well demonstrated during infections caused by Gram-negative bacteria, such as *Pseudomonas* sp., *E. coli*, *Salmonella typhimurium*, *Shigella* sp., *Klebsiella pneumoniae*, and *Enterobacter* sp. [1,15,16,17]. However, little is known about Gram-positive infections. For example, in human patients with meningitis provoked by *Streptococcus pneumoniae*, the BPI concentration increased significantly, suggesting a broad role during infection. BPI binds to bacterial wall molecules such as lipopeptides and lipoteichoic acids in vitro, and these interactions increase the TNF-α production in peripheral blood mononuclear cells (PBMCs) [18]. Furthermore, the N-terminal fragment of 21-kDa of BPI (rBPI21) binds to *S. pneumoniae* and increases the apoptotic response. rBPI21 peptide enhances the association of pneumococci with macrophages and confers an essential survival advantage on TLR4-defective mice after mucosal pneumococcal exposition [19].

Respiratory diseases represent a major global problem; infectious diseases cause millions of deaths worldwide [20]. Cystic fibrosis and chronic obstructive pulmonary disease are respiratory conditions characterized by persistent infection commonly generated by *Pseudomonas aeruginosa*. This infection contributes to the production of autoantibodies against BPI (anti-BPI), decreasing the BPI concentration in serum and the ability to kill the bacteria and inducing an exacerbated inflammatory response [21]. Pulmonary tuberculosis is caused by *Mycobacterium tuberculosis*, a very successful Gram-positive bacterium. This pathogen grows actively or remains dormant for years within its host [22]. Tuberculosis is a health problem in the world. A quarter of the world’s population is infected, and 5–10% will develop the active disease [23]. Efforts to contain tuberculosis have been restricted by the lack of an effective vaccine, as well as the high incidence of multi-drug-resistant strains. Therefore, it is essential to have various strategies to treat this infection, such as new molecules with diverse antimicrobial and immune regulatory activities, without having as many adverse effects as those that commonly occur with anti-tuberculosis therapy. 

Recently, it was demonstrated that the BPI gene expression significantly decreased during infection in a model of human macrophages infected with *M. tuberculosis* H37, the virulent strain. However, when the infected macrophages were treated with immunomodulatory lipids such as resolvin D1 (RVD1) and maresin 1 (MaR1), the BPI expression increased, the TNF-α production was reduced, and the death of mycobacteria was favored [24]. These findings suggested the association of the increase in BPI with mycobacterial death in human macrophages. Therefore, the main objective of this work was to determine the bactericidal activity of BPI against *M. tuberculosis* in vitro and within the context of an infection.

## 2. Materials and Methods

### 2.1. Strains and Culture Conditions

The *M. tuberculosis* H37 Ra (avirulent) and H37 Rv (virulent) strains were obtained from the American Type Culture Collection (25177 and 27294, ATCC, Rockville, MD, USA). We transformed avirulent *M. tuberculosis* H37 with the pCherry8 plasmid (plasmid number 24663, provided by Addgene, Cambridge, MA, USA). This strain constitutively expresses the fluorescent reporter protein mCherry from the Psmyc promoter [25].

Bacterial stocks were maintained at −70 °C, subsequently thawed, and were grown in Middlebrook 7H9 liquid medium supplemented with 10% ADC (Beckton Dickinson, BD, San Jose, CA, USA) and 0.2% glycerol (7H9c), incubated at 37 °C with constant shaking (120 rpm) until the early exponential phase (7 days). CFU counting was performed on Middlebrook 7H10 agar medium (Difco, Detroit, MI, USA) supplemented with 10% OADC (BD) and 0.05% glycerol (7H10c) and incubated for 21 days at 37 °C.

Before macrophage infection, mycobacterial strains were disaggregated to reach an individual cell suspension. For this purpose, the bacterial stock was thawed and centrifuged at 8000 rpm for 5 min and washed in RPMI medium (Lonza, Walkersville, MD, USA) without antibiotics. The bacterial suspension was homogenized with 3 mm diameter glass beads in a vortex for 5 min, and then the suspension was centrifuged at 2000 rpm for 2 min as described previously [26]. 

### 2.2. Cell Culture and Differentiation

The human peripheral blood mononuclear cells (PBMCs) were isolated from buffy coats from healthy donors to the blood bank. The Institutional Review Board approved the protocol. The blood was diluted 1:4 with RPMI 1640 (Lonza) and centrifuged over Lymphoprep solution (Stemcell Technologies, Vancouver, BC, Canada) to obtain PBMCs. The monocytes were isolated from the PBMCs through positive selection using antibodies coupled to magnetic microbeads (Miltenyi Biotech, Auburn, CA, USA). To obtain monocyte-derived macrophages (MDMs), monocytes were cultivated for 7 days in RPMI supplemented with 200 mM L-glutamine (Lonza) and 10% heat-inactivated human serum (Valley Biomedical, VA, USA) at 37 °C in 5% CO_2_.

The THP1 cell line was obtained from ATCC and maintained at 0.2–1.0 × 10^6^ cells/mL in RPMI medium supplemented with 2 mM L-glutamine, 10 mM HEPES buffer, 1 mM sodium pyruvate (Lonza), 10% heat-inactivated fetal bovine serum (Hyclone, USA), and 50 μM β-2-mercaptoethanol (Bio-Rad Laboratories, Hercules, CA, USA) in 75 cm^2^ bottles in 5% CO_2_ at 37 °C. Cells were differentiated into macrophages by adding 50 nM of phorbol-12-myristate-13-acetate (PMA, Sigma-Aldrich, St. Louis, MO, USA) for 72 h, herein termed THP-M.

The viability of both cell types was counted in the Neubauer chamber stained with trypan blue (Lonza), a non-vital dye.

### 2.3. Internalization of BPI

#### 2.3.1. Western Blot

To assess the internalization of BPI, 3 × 10^6^ macrophages per well were incubated with rBPI (recombinant human BPI Fc chimera protein; R&D Systems, Minneapolis, MN, USA) at 0, 1, 5, and 10 µg per well for 1 h. Cell lysates were prepared, and proteins were resolved via 10% SDS-PAGE and transferred to a PVDF membrane. The blots were incubated with a human anti-BPI primary antibody (R&D Systems) followed by an anti-mouse HRP secondary antibody (DSHB, University of Iowa, Iowa City, IA, USA). Immunoblots were visualized with Luminol reagent (Bio-Rad), and densitometric quantification of blots was performed using the Image Lab software version 5 (Bio-Rad). We detected rBPI at 77.3 kDa and endogenous protein at 52 kDa. 

#### 2.3.2. Fluorescence Microscopy

Cells were cultured in eight-well Lab-Tek II chambers at 0.15 × 10^6^ cells/well and differentiated into macrophages. Then, macrophages were infected with mycobacteria (MOI 1:10), treated with 200 ng of rBPI, and incubated for 1 h at 37 °C in 5% CO_2_. After discarding non-phagocytosed bacteria, the cells were incubated for 24 h with the same amount of rBPI. Uninfected cells treated with rBPI were included. The cells were fixed with 4% paraformaldehyde (Sigma-Aldrich) and incubated with an anti-hBPI mouse antibody (R&D Systems) and anti-mouse coupled with Alexa Fluor 488 rabbit antibody (Thermo Fisher Scientific, Waltham, MA, USA). Counter-staining with Hoechst (Enzo Life Sciences, Farmingdale, NY, USA) was used to detect nuclei. Cells were visualized under a fluorescence AxioScope.A1 microscope (Carl Zeiss, Oberkochen, DEU), and images were acquired and analyzed with the ZEN Pro software v. 2012 (blue edition) (Carl Zeiss). We analyzed the infected MDM and THP-M cell preparations in a confocal microscope to confirm the intracellular localization of mycobacteria. Images were acquired in an Olympus FV1000-IX81 microscope with the FV10-ASW v.4 software and analyzed with the Fiji software v. 1.54i [27]. Orthogonal projections of confocal sections showed bacteria inside the cell (Appendix A). 

### 2.4. Intracellular Antimicrobial Activity of BPI

Macrophages were infected with avirulent or virulent *M. tuberculosis* H37 at an MOI of 5 and incubated for 1 h at 37 °C in 5% CO_2_ with 5, 10, or 20 µg/mL; then, the cells were washed three times with RPMI to discard non-phagocytosed bacteria. The medium and rBPI were refreshed, and macrophages were further incubated at 37 °C in 5% CO_2_ for 72 h.

After incubation, supernatants were discarded, and cells were lysed with 0.1% SDS for 10 min and then neutralized with 10% bovine serum albumin (BSA, Sigma Aldrich). Lysates were serially diluted and plated on 7H10c agar plates in triplicate. The colony-forming units (CFUs) were counted after 21 days.

### 2.5. In Vitro Antimicrobial Activity of BPI

The broth microdilution susceptibility assay was used to detect the antimicrobial activity of rBPI against avirulent and virulent strains of *M. tuberculosis* H37 [28]. The vials of frozen bacteria were thawed and cultured in a 7H9c liquid medium for 7 days until the exponential phase. The culture was transferred to 96-well plates (10^5^ bacteria/mL) and incubated for 96 h in the presence of 5, 10, or 20 µg/mL of rBPI. The samples were transferred to Eppendorf tubes, and the bacteria were disaggregated, serially diluted, and seeded in triplicates on 7H10 agar plates. The CFUs were quantified after 21 days. Untreated bacteria and treatment with rifampicin were used as controls.

### 2.6. BPI’s Effect on TNF-α

MDMs and THP-Ms were infected with 100 μL of the previously made mixture of rBPI, bacteria were disaggregated according to the infection index (MOI 1:5) and incubated for 5 min to promote phagocytosis, and 150 μL of RPMI medium was immediately added and incubated again for 1 h at 37 °C. Subsequently, cells were washed three times with 300 μL of RPMI medium without antibiotics. Finally, 100 μL of the corresponding BPI solution (1, 5, and 10 µg/mL) and 200 μL of RPMI medium were added and incubated for 24 h. The supernatant was collected and stored at −70 °C until TNF-α detection with ELISA [29]. The value obtained from duplicates was reported as a percentage of the control; we set the TNF-α production from infected cells that did not receive BPI treatment at 100%.

### 2.7. Statistical Analyses

All data were analyzed using the Shapiro–Wilk test for normal distributions. Data with a normal distribution were analyzed using the parametric RM one-way ANOVA with the Geisser–Green–House correction test followed by Tukey’s test for multiple comparisons. Statistical significance for *p*-values was set at <0.05. The TNF-α results were analyzed with Wilcoxon’s one-sample *t*-test to determine whether they differed from 100%. All statistical analyses were run using GraphPad Prism version 9.5.0 (San Diego, CA, USA).

## 3. Results

### 3.1. Internalization of BPI into Human Macrophages

Before assessing the bactericidal activity of BPI against intracellular mycobacteria, we first demonstrated that human recombinant BPI is internalized into uninfected and infected macrophages. Furthermore, we found no detectable levels of the native BPI with a specific band at approximately 52 kDa in any condition in both types of cells. We observed that BPI is rapidly internalized into uninfected cells, since the protein was detected intracellularly within one hour. MDMs internalized BPI in a concentration-dependent manner with 1.3-, 7.0-, and 11.6-fold increases at 1, 5, and 10 µg/mL, respectively. At the same time, THP-Ms incorporated less BPI; the internalization at 1, 5, and 10 µg/mL was 1.0, 3.9, and 5.1, respectively (Figure 1).

Furthermore, fluorescence microscopy images corroborated the intracellular localization of BPI remaining after 24 h in uninfected and infected human macrophages. We visualized the cells in phase contrast and identified the nucleus with Hoechst (ex405/em450 nm λ); BPI was identified using a secondary antibody coupled with Alexa Fluor 488 (ex494/em517 nm λ), and avirulent *M. tuberculosis* H37 constitutively expressing mCherry protein was detected at ex587/em610 nm λ. We merged the images and quantified the percentage of cells that internalized BPI. We observed that the uninfected cells incorporated 39 and 25% in MDMs and THP-Ms, while the infected cells incorporated 31 and 41% in MDMs and THP-Ms, respectively (Figure 2A–D).

### 3.2. BPI Inhibits the Intracellular Growth of M. tuberculosis in Macrophages

Cells treated with BPI significantly decreased the mycobacterial burden in MDMs and THP-Ms relative to untreated cells after 3 days of incubation (Figure 3). The reductions in intracellular virulent bacteria were 39% and 8% in MDMs and 33% and 19% in THP-Ms when incubated with 10 and 20 µg/mL BPI, respectively. In contrast, 10 and 20 µg/mL BPI decreased the intracellular avirulent bacteria by 37% and 11% in MDMs and 51% and 25% in THP-Ms. However, BPI did not completely inhibit intracellular bacterial growth like rifampicin, which kills mycobacteria at 200 ng/mL.

### 3.3. BPI Shows Inhibitory Activity against M. tuberculosis

We determined the bactericidal effect of recombinant BPI on mycobacterial strains using the microdilution assay. We observed that different concentrations of this protein partially inhibited bacterial growth in both strains (Figure 4). The reductions in the virulent growth was 25%, 35%, and 36% with 5, 10, and 20 μg/mL BPI, respectively. In contrast, 5, 10, and 20 μg/mL BPI decreased the growth of the avirulent strain by 25%, 35%, and 34%, respectively. Concentrations higher than 20 µg/mL did not significantly decrease mycobacterial growth. In contrast, rifampicin at 200 ng/mL, the bactericidal antibiotic used as a control, decreased the microbial growth by 100% in both strains.

### 3.4. BPI Inhibits TNF-α Production

Previous studies have shown that mycobacterial infection in vitro exacerbates TNF-α production [30]. Our data showed that infected cells produced TNF-α (48.5 ± 23 ng/mL in MDMs and 187 ± 18 ng/mL in THP-Ms) 24 h post-infection. The rBPI treatment (1, 5, and 10 mg/mL) significantly reduced the TNF-α secretion (*p* < 0.05). MDMs showed 52%, 43%, and 24% of production compared to the control, while THP-Ms presented 57%, 50%, and 4% of production relative to the control (100%), (Figure 5).

## 4. Discussion

*M. tuberculosis* is an intracellular parasite that mainly causes pulmonary infection in individuals with weakened immune systems, children, and the elderly, and it is difficult to eradicate [31]. This study aimed to determine the contribution of exogenous BPI to macrophages’ control of the intracellular growth of *M. tuberculosis*. First, we analyzed the uptake of exogenous BPI by macrophages. We observed the rapid internalization of exogenous BPI by human macrophages derived from monocytes and the macrophages derived from the THP-1 monocytic cell line. Previous studies reported that biotinylated BPI is quickly incorporated into the THP-1 cell line through pinocytosis in a concentration-independent fashion [32]. In our experiments, the uptake of rBPI in THP-Ms was also independent of concentration after one hour of treatment. However, in MDMs, the protein incorporation was dependent on the concentration and twice larger than in THP-Ms. Probably, macrophages may immediately activate diverse mechanisms for BPI internalization, such as macropinocytosis or clathrin-mediated endocytosis, lipid-raft-mediated endocytosis, caveola-mediated endocytosis, or direct membrane penetration through pore or micelle formation, mechanisms involved in protein transportation into human cells [33]. Although MDMs incorporated more rBPI than THP-Ms after one hour of treatment, both cell types showed similar percentages of the remaining intracellular BPI at 24 h, regardless of infection. Human BPI is mainly produced by neutrophils for bactericidal function; it is possible that it can be internalized by macrophages to control intracellular bacteria [34].

*M. tuberculosis* may be opsonized with BPI, facilitating the internalization of the bacteria into macrophages through several mechanisms. For instance, the opsonophagocytosis of *E. coli* by BPI depends on the C-terminal region, which increases bacterial surface hydrophobicity and, thereby, promotes phagocytosis. This phenomenon is independent of the complement receptor 3 (CR3) protein and the CD14 receptor [3]. BPI may favor phagocytosis, promote bacterial clearance and intracellular disassembly, and generate ultimate mycobacterial elimination. BPI has also been reported in neutrophil extracellular traps, providing another path for bacterial killing and clearance [5,35].

Next, we examined the inhibitory effect of BPI on intracellular bacterial growth. We found that recombinant BPI significantly reduced the intracellular growth of avirulent and virulent strains of *M. tuberculosis* H37 without affecting macrophage viability, although the control was partial. The internalized BPI probably exerts bactericidal effects directly on the mycobacteria or indirectly by modulating the macrophage responses. In Gram-negative bacteria, BPI binds to their LPSs; the N-terminal domain of BPI bound at the bacterial surface destabilizes the integrity of the membrane, leading to bacterial lysis and cell death [18,36,37,38]. Interestingly, BPI also binds Gram-positive bacteria and boosts immune response; rBPI21 peptide (the N-terminal fraction of 21 amino acids) binds to live *S. pneumoniae* and enhances the association with mouse macrophages [19]. The complete BPI binds *Staphylococcus aureus* lipopeptides and inhibits the growth of the L-form of *Staphylococcus aureus* and *Streptococcus pyogenes* [39]. The interaction between BPI and lipopeptides is blocked by LPSs, suggesting that the N-terminal domain interacts with other Gram-positive bacteria [18]. Recently, it was described that the recombinant BPI of fish *Trachinotus ovatus* (toBPI), which is homologous to the human protein, efficiently inhibits the growth of Gram-positive bacteria such as *Streptococcus agalactia* and *Streptococcus iniae*. Also, toBPI increases bacterial cell membrane permeability like human BPI in Gram-negative bacteria, highlighting the essential role of this protein during infectious processes in different animal species [40]. Furthermore, in an infected human granulocyte model, influenza A virus variants such as H1N1, H3N2, and H5N induce the release of BPI. In the same model, the rBPI_21_ peptide inhibits viral infectivity. It changes the viral structure of the H1N1 strain while significantly diminishing interferon alpha (IFN-α) and IL-6 production in a concentration-dependent manner [41]. This evidence suggests that BPI is a molecular pattern recognition molecule associated with different types of pathogens. 

The mycobactericidal action of BPI may damage the bacterial surface and partially increase membrane permeability, showing a bacteriostatic effect. LPSs and lipoarabinomannans (LAMs) have similar chemical structures; LAM is a glycolipid from the cell wall of *Mycobacterium* and shares an affinity with other LPS-interacting proteins, such as LBP. LAMs can also bind to the CD14 receptor in the same manner as that of LPSs [42]. Thereby, BPI may interact with the LAMs of mycobacteria, favoring phagocytosis as in Gram-negative bacteria. We observed the killing of the mycobacteria inside the macrophages and confirmed the direct bactericidal effect in macrophage-free bacterial cultures. Further studies are needed to better characterize the bactericidal role of BPI in macrophages infected with mycobacteria.

Lastly, we investigated the regulatory role of BPI in the macrophage responses. Our data showed a regulatory effect of BPI on the TNF-α production in the infected macrophages. The mycobacterial infection in vitro causes exacerbated induction of TNF-α [30]. Excessive production of this cytokine induces reactive oxygen species (ROSs) in infected macrophages. Although the early stage of infection is essential to increasing macrophage microbicidal activity, ROSs rapidly induce programmed necrosis. This releases bacteria, generating extracellular growth, spreading the infection to other organs, and aggravating the disease [30]. Furthermore, the pathogenic effect of TNF-α has been well documented since overexpression can cause damage to multiple organs, causing lung dysfunction during tuberculosis [43]. Here, we found that BPI decreased the TNF-α production in infected macrophages. BPI may bind and reduce the number of intracellular mycobacteria, helping to regulate the innate immune response. BPI may interact directly with the mycobacterial surface to limit the cytokine production that follows the recognition of mycobacteria, thus inducing an anti-inflammatory effect, as reported for Gram-negative bacterial infections [44]. This work suggests that BPI plays an essential role during mycobacterial infection; it exerts direct and indirect mechanisms through the opsonization of bacteria, the activation of related pathways, and the counter-regulation of proinflammatory cytokines such as TNF-α. Another model of human macrophages infected with *M. tuberculosis* revealed that BPI gene expression is significantly reduced during infection. This effect could be a strategy of the bacteria to escape death, promoting their persistence and, thus, remaining in a state of latency for a longer time. However, the expression of BPI in macrophages infected with *M. tuberculosis* can be induced, favoring the death of mycobacteria [24]. Here, we also demonstrated that the addition of exogenous BPI contributes to the death of mycobacteria, although the precise mechanisms are unknown.

## 5. Conclusions

Recombinant BPI is internalized by macrophages and contributes to controlling the intracellular growth of *M. tuberculosis* in avirulent and virulent strains. BPI likely inhibits bacterial growth by binding to the cell surface. Furthermore, BPI reduced TNF-α expression in human macrophages at 24 h. We concluded that BPI has a dual effect: a direct bactericidal effect and indirect control by regulating macrophage immune responses.

## Figures and Tables

**Figure 1 biomolecules-14-00475-f001:**
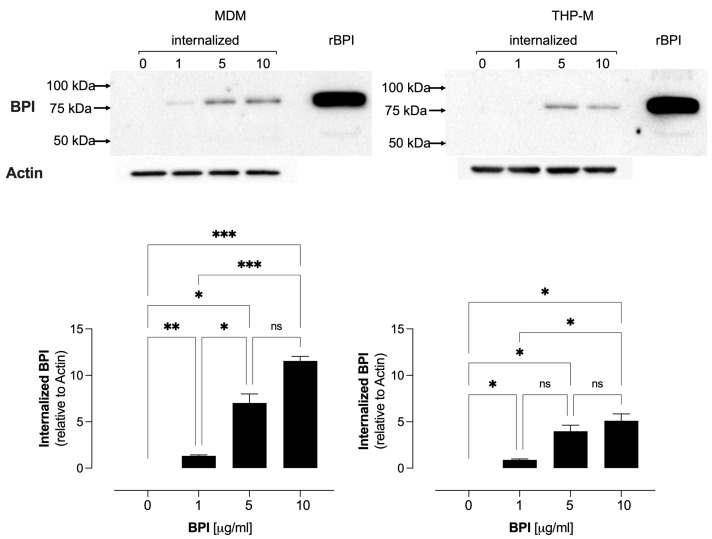
Internalization of human recombinant BPI (77.3 kDa) into human macrophages (MDMs and THP-Ms) according to a Western blot (original images can be found in Appendix A). The BPI internalization was calculated relative to the actin ratio. The data represent the mean ± SEM (*n* = 4); ns, not significant, * *p* < 0.05, ** *p* < 0.01, and *** *p* < 0.002.

**Figure 2 biomolecules-14-00475-f002:**
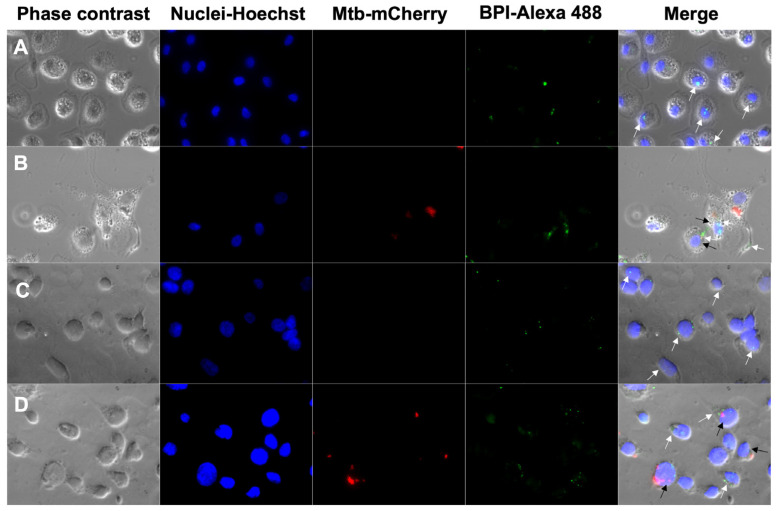
Internalization of BPI into uninfected (**A**,**C**) and infected (**B**,**D**) human macrophages (MDMs and THP-Ms). Cells were infected with *M. tuberculosis* (Mtb) and then treated with BPI (200 ng/mL) for 24 h. (**A**) BPI was detected via fluorescence microscopy using an anti-human BPI antibody and then a secondary antibody coupled with Alexa Fluor 488. *M. tuberculosis* was detected according to the expression of *mCherry* and nuclei with the Hoechst stain. The merged images show Mtb-mCherry-positive cells (black arrow) and BPI-positive cells (white arrow). Images are representative of three independent experiments; images were acquired at 100×.

**Figure 3 biomolecules-14-00475-f003:**
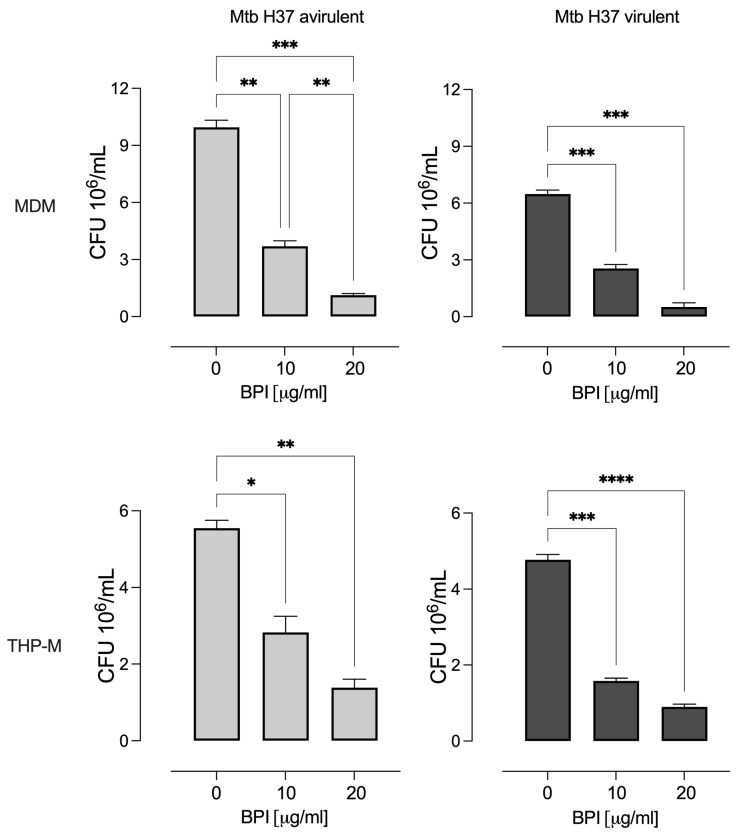
Inhibitory effect of human recombinant BPI on the intracellular growth of *M. tuberculosis*. Human macrophages (MDMs and THP-Ms) were infected with avirulent and virulent strains of *M. tuberculosis* H37 (Mtb) for 3 days. The cells were lysed, and the intracellular bacteria were serially diluted and plated in 7H10 medium. The CFUs were counted after 21 days. The data represent the mean ± SEM (*n* = 4); * *p* < 0.05, ** *p* < 0.01, *** *p* < 0.002, and **** *p* < 0.001.

**Figure 4 biomolecules-14-00475-f004:**
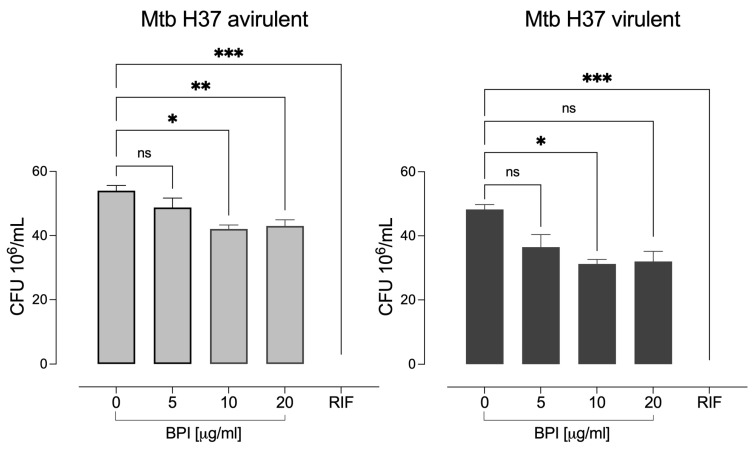
Inhibitory effect of human recombinant BPI on the growth of *M. tuberculosis*. Avirulent Mtb H37 and virulent Mtb H37 strains were cultivated in 7H9c Middlebrook medium supplemented with different concentrations of BPI for 96 h. Surviving bacteria were plated, and CFUs were counted after 21 days. The data represent the mean ± SEM (*n* = 4); ns, not significant, * *p* < 0.05, ** *p* < 0.002, and *** *p* < 0.001.

**Figure 5 biomolecules-14-00475-f005:**
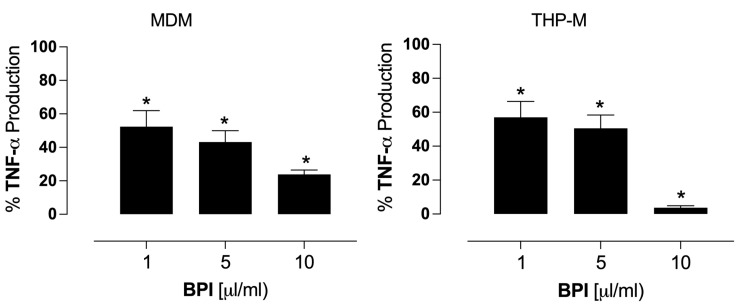
Inhibitory effects of BPI on the TNF-α production in infected cells. Macrophages were infected with avirulent *M. tuberculosis* H37 and incubated for 1 h. Then, the macrophages were washed, and BPI was added and incubated for 24 h. The TNF-α production was quantified with ELISA. The TNF-α production of macrophages infected but not treated with BPI used here as a control was set to 100%. The data represent the percentual average relative to the control ± SEM (MDM, *n* = 6 and THP-M, *n* = 4); * *p* < 0.05.

## Data Availability

The data supporting this study’s findings are available upon reasonable request from the corresponding author.

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
