# Peer review of "Bactericidal Permeability-Increasing Protein (BPI) Inhibits Mycobacterium tuberculosis Growth"

_biomolecules, 2024, doi:10.3390/biom14040475_

Round 1

Reviewer 1 Report

Comments and Suggestions for Authors

The manuscript titled " The bactericidal permeability-increasing protein (BPI) inhibits Mycobacterium tuberculosis growth” valuated the antimycobacterial and immunoregulatory properties of BPI in Mycobacterium tuberculosis-infected human macrophages. In general, manuscript is organized and well written other than some mistakes and formatting errors indicated below.

1.       Section 2.3, how to distinguish rBPI and the BPI expressed by the cells themselves.

2.       Figure 2, how to distinguish intracellular Mtb and the Mtb that attached to the cytomembrane.

3.       Section, how to confirm that the non-phagocytosed bacteria was removed completely.

4.       Figure 4, error bar of control

5.       SEM or SD, please confirm.

Author Response

RESPONSE TO REVIEWERS

We appreciate all the reviewers' comments and suggestions. Addressing the reviewers' concerns greatly improved our manuscript. Such improvements include new information in the introduction and the discussion sections, although no reviewer requested them. We even included an additional figure with results that strengthen our conclusions (Fig.4). Please find below the point-by-point response to all the reviewers’ comments.

REVIEWER 1

The manuscript titled " The bactericidal permeability-increasing protein (BPI) inhibits Mycobacterium tuberculosis growth” valuated the antimycobacterial and immunoregulatory properties of BPI in Mycobacterium tuberculosis-infected human macrophages. In general, manuscript is organized and well written other than some mistakes and formatting errors indicated below.

  1. Section 2.3, how to distinguish rBPI and the BPI expressed by the cells themselves.

A1: In the experimental conditions we used, we found no detectable levels of the native BPI in our macrophages (see the figure below). The recombinant BPI was absent in the lane labeled “0” because no rBPI was added to the cell culture, and the antibody did not detect the native protein (Fig. 1, upper left panel and upper right panels). The revised version of the manuscript indicates in the method section (lines 132-133) the molecular weight of the rBPI (77.3 kDa), and a note on the absence of native BPI (52 kDa) detection appears in the results section (Fig. 1). Also, we included the molecular weight annotation indicating the location of the rBPI and the place where the native protein should appear (lines 190-191 and 198).

Figure 1. Human recombinant BPI (77.3 kDa) internalization into human macrophages MDM and THP-M by western blot. The BPI internalization was calculated relative to the actin ratio. The data represent the mean ± SEM (n = 4); ns, not significant, **p<0.05, ** p<0.01, and *** p<0.002.

  1. Figure 2, how to distinguish intracellular Mtb and the Mtb that attached to the cytomembrane.

A2: To address this issue, we reanalyzed the cell preparations in a confocal microscope and prepared a z-stack image to demonstrate the internalization of the Mtb. The image appears in the revised version of the manuscript as Supplementary Figure 1 and is commented on in the methods section.

  1. Section, how to confirm that the non-phagocytosed bacteria was removed completely.

A3: Despite the extensive washing to remove the non-phagocytosed bacteria, some extracellular Mtb remained in the slide outside of cells that we could not completely remove. However, the image analysis focused on the cells, and we are confident that the bacteria were intracellularly localized. The confocal analysis demonstrated the subcellular localization of Mtb (Supplementary Figure 1), which indirectly confirms that our analysis focused on intracellular Mtb.

  1. Figure 4,error bar of control

A4: Now Figure 4 is Figure 5. The control was set to 100% to normalize individual responses. The results in terms of pg/mL remain from the original version of the manuscript in the results section (section 3.4). Since all data values were 100, no error bars were produced. However, to prevent confusion, we removed the control bar and indicated in the figure legend what the response percentage refers to.

  1. SEM or SD, please confirm.

A5: All figure legends state the depicted parameters. Mean ±SEM is indicated where appropriate.

Reviewer 2 Report

Comments and Suggestions for Authors

Aticles presentated by Gonzalez et al., shows interesting study on BPI. I suggest change the following things in manuscript.

correct the substitles and reference section. As number and patterns found inconssitance in both cases.

Author Response

RESPONSE TO REVIEWERS

We appreciate all the reviewers' comments and suggestions. Addressing the reviewers' concerns greatly improved our manuscript. Such improvements include new information in the introduction and the discussion sections, although no reviewer requested them. We even included an additional figure with results that strengthen our conclusions (Fig.4). Please find below the point-by-point response to all the reviewers’ comments.

REVIEWER 2

Articles presented by Gonzalez et al., shows interesting study on BPI. I suggest change the following things in manuscript.

Correct the subtitles and reference section. As number and patterns found inconsistence in both cases.

A: We appreciate the suggestion. The subtitles and all the sections were carefully revised.

Reviewer 3 Report

Comments and Suggestions for Authors

The reviewer would like to thank the authors for their submission titled, “The bactericidal permeability-increasing protein (BPI) inhibits Mycobacterium tuberculosis growth.”  In this article, the authors hypothesize that internalization of the BPI will inhibit growth of M. tuberculosis through internalization into macrophages.  They address this hypothesis through antimicrobial assays.

The reviewer will provide a critique of both the Stylistic and Technical aspects of the work:

Style:

1.      There may be some formatting issue with Line 176, it appears to be cut off.

2.      In Figure 2, it appears there may be one cell that exhibits both internalized BPI and Mbt, it may be worth highlighting this for the readers using an arrow, circle, etc.

3.      In Line 190 the authors begin talking about virulent and avirulent strains.  It might be helpful, back on Line 68, just to explicitly state that H37Ra is the avirulent strain and H37Rv is the virulent strain as this might not be obvious to all readers.

4.      In Figure 3, the authors denote significance by saying in the caption, “*p<0.05, and *** p<0.001.” However, the figure also appears to contain the notations “ ** ” and “ **** ”.  Could the authors please clarify in the caption what is meant by the latter two?

Technical

1.      Could the authors please provide in line citations to their microbial assays.  A standardized assay such as CLSI or EUCAST might be most helpful for the readers.

2.      To further probe the mechanism of internalization did the authors consider incubating macrophages with BPI at 25 C?  This can be utilized to limit endocytosis and may provide insights into the route of delivery.

Author Response

RESPONSE TO REVIEWERS

We appreciate all the reviewers' comments and suggestions. Addressing the reviewers' concerns greatly improved our manuscript. Such improvements include new information in the introduction and the discussion sections, although no reviewer requested them. We even included an additional figure with results that strengthen our conclusions (Fig.4). Please find below the point-by-point response to all the reviewers’ comments.

REVIEWER 3

The reviewer would like to thank the authors for their submission titled, “The bactericidal permeability-increasing protein (BPI) inhibits Mycobacterium tuberculosis growth.”  In this article, the authors hypothesize that internalization of the BPI will inhibit growth of M. tuberculosis through internalization into macrophages. They address this hypothesis through antimicrobial assays.

The reviewer will provide a critique of both the Stylistic and Technical aspects of the work 

Style:

  1. There may be some formatting issue with Line 176, it appears to be cut off.

A1: We appreciate the suggestion. We have revised the text (lines 207-209 of the revised version of the manuscript).

  1. In Figure 2, it appears there may be one cell that exhibits both internalized BPI and Mtb, it may be worth highlighting this for the readers using an arrow, circle, etc.

A2: The figure has been amended. We used arrows to point to the localization of BPI and Mtb, which we explained in the figure legend.

  1. In Line 190 the authors begin talking about virulent and avirulent strains.  It might be helpful, back on Line 68, just to explicitly state that H37Ra is the avirulent strain and H37Rv is the virulent strain as this might not be obvious to all readers.

A3: For clarity, we introduced the virulence of each strain when mentioned. The entire manuscript was modified in this respect.

  1. In Figure 3, the authors denote significance by saying in the caption, “*p<0.05, and *** p<0.001.” However, the figure also appears to contain the notations “ ** ” and “ **** ”.  Could the authors please clarify in the caption what is meant by the latter two?

A4: We used GraphPad Prism; the analysis software uses *p<0.05, **p<0.01, ***p<0.002, and ****p<0.0001 to annotate statistical significance. We revised the figure legends and amended them accordingly.

Round 2

Reviewer 1 Report

Comments and Suggestions for Authors

No further comment